# Stronger Together? Determinants of Cooperation Patterns of Migrant Organizations in Germany

Nils Friedrichs *[iD] and Marie Mualem

The Expert Council on Integration and Migration (SVR), Neue Promenade 6, 10178 Berlin, Germany;
mualem-schroeder@svr-migration.de
* Correspondence: friedrichs@svr-migration.de

**Abstract:** Based on a cross-regional survey of 764 migrant organizations (MOs) in Germany, this study examines the relationship between the content-related orientation (activity areas) of a migrant organization, its human and financial resources, and its cooperation patterns, e.g., the tendency to become a member of umbrella associations and to cooperate with various actors in society. Resources as well as activity areas of an MO have effects on such collaborations, but they differ depending on the cooperation partner. MOs with a focus on social work are more likely to cooperate with public institutions and civil society organizations and have a larger network overall than MOs active in other areas. In addition, human resources are more important for cooperation than funding. While paid employees increase the probability of cooperation with public institutions as well as the membership in an umbrella association, the number of volunteers correlates positively with the tendency to cooperate with other organizations of civil society and religious communities. In particular, a broad base of paid and volunteer staff increases the chances of cooperation.

**Keywords:** migrant organizations; social capital; social protection; welfare; networks; cooperation





## 1. Introduction—Migrant Organizations and Cooperation Structures

In 2021, more than a quarter of the population in Germany (27.2%) had a migration background (Statistisches Bundesamt 2022, p. 39), which means that a person or at least one of their parents did not have German citizenship by birth (Statistisches Bundesamt 2022, p. 22). Accordingly, the need for diversity-sensitive services has been high for years and continues to grow, especially in the field of social work.[1] This field includes intercultural educational work, migration social work, the cultural opening of welfare work or the integration of refugees (Huth 2019, p. 2). Migrant organizations[2] are increasingly emerging as central actors in the field of social work themselves.

In this article, the term migrant organization (MO) refers to non-profit organizations, in which at least half of the people involved, and (if applicable) half of the employees have a migration background; for which a migration experience in the broadest sense forms the common ground to understand the organizations' main goals and activity fields.[3] In Germany, growing awareness with regard to the potential of MOs in the field of diversity-sensitive social work is one reason for a substantially increased political interest in their work, especially since the mid-1990s and particularly since the introduction of the Immigration Act in 2005 (Halm 2013, p. 20f).

Several studies (see, for example, Klöckner 2016; Priemer and Schmidt 2019; SVR-Forschungsbereich 2020a) have shown that MOs are active in almost all areas of civic engagement. At the same time, however, clear focal points of their work in the social sphere can be identified in the social sector and social services (Predelli 2008; Reinecke et al. 2010; Priemer and Schmidt 2018; SVR-Forschungsbereich 2020a; Góis and Marques 2023), which is particularly true for MOs in Germany.[4] The strong involvement of migrant organizations in the social sector in Germany continues a tradition that goes back to the

1970s, when families increasingly joined the recruited foreign workers from abroad (Hunger 2004, p. 10f).[5] One political strategy to strengthen the work of migrant organizations has been to promote cooperation and networking between them, e.g., via funding and support for the establishment of umbrella organizations, especially in the past ten years. The promotion of umbrella associations is also explained by the political desire to obtain a better overview of the social commitment of migrant organizations, and to gain access to their expertise (Mualem Sultan 2022). MOs, on the other hand, benefit from cooperation in two respects: first, cooperation increases MOs' efficiency in their activities through sharing knowledge and resources with other organizations in civil society[6], as well as with public institutions (Huth 2011; Fauser 2016). Second, if MOs' activities focus on participation and integration in a society, cooperation might be a precondition to meet their objectives of building bridges in society (Odmalm 2004, p. 473; SVR-Forschungsbereich 2020a, p. 53). Thus, the need for cooperation seems to depend, on the one hand, on MOs' targets. On the other hand, studies show that human and financial resources influence the possibility and success of MOs' collaborations (Hunger and Metzger 2011; Huth 2011; Halm et al. 2020; Klie 2022).

Although recent studies show that most MOs maintain cooperation (Predelli 2008; SVR-Forschungsbereich 2020a; Halm et al. 2020; Klie 2022), pro-cooperation arguments should be particularly important for the MOs that offer services in the field of social work. Social work is often demanding or even requires special skills and qualifications. This might result in an increased need for advice, qualification and support between migrant organizations with a particular focus on social work. Against this background, cooperation might be even more important to reach longer-term goals. Those who do not act as lonesome warriors thus also increase their chances of gaining all the skills that are required for being officially recognized as a social work organization (Träger der sozialen Arbeit) in the long term.

These descriptions already make it clear that, in principle, many different types of cooperation are conceivable. In their study on MOs in Germany, Hunger and Metzger (2011), for instance, distinguish between six different forms of cooperation (p. 11).[7] Furthermore, cooperation can also be differentiated with regard to the partners involved. The type of cooperation, as well as the specific cooperation partners, might determine whether or not a collaboration emerges and how successful it can be considered. Regarding cooperation between MOs, some studies indicate that organizations, which share the same ideological goals and target groups tend to compete (e.g., for funding) rather than collaborate with each other, while complementary organizations prefer cooperation to reach their goals (Vermeulen 2013, p. 460; Vermeulen and Brünger 2014, p. 992). Using synergies is also likely to be the aim of collaborations between MOs and other non-migrant civil organizations. However, in the past (e.g., in so-called "tandem projects"), a problem would sometimes arise in which migrant organizations felt that they did not benefit from a cooperation in the same way as their partners. While academia, politics, and society normally perceive the idea of cooperation as something positive, it would be naïve to assume that collaborations are always and automatically profitable for both sides (Kumbruck 2001). It is by no means guaranteed that cooperation will always take place at eye level and to mutual advantage; this has also been highlighted in studies on migrant organizations in Germany and other countries (Hunger and Metzger 2011; Weiss 2013; Fauser 2016).

Following these general considerations, the article examines the conditions that determine whether MOs enter into cooperative agreements with different partners. In doing so, we focus firstly on the importance of the organization's activity focus, and secondly on its resource endowment. The central research question is therefore: with regard to MOs in Germany, how are their activity fields and their human and financial resources related to the establishment of cooperation?

The article is divided into the following sections: in Section 2, we discuss theoretical considerations of the social capital theory by Robert D. Putnam (1995, 2020) and provide an overview of recent studies that deal with the conditions for cooperation among MOs.

We then formulate our hypotheses. After describing the data and indicators used in our study (Section 3), we test these hypotheses and discuss our results in Section 4. The article concludes with a discussion of the limitations of our analyses (Section 5).

## 2. Migrant Organizations and Cooperation—Theoretical Considerations and Empirical Findings

In this section, we refer to theoretical considerations and discuss the state of empirical knowledge on the importance of the activity fields of MOs (Section 2.1), as well as on the relevance of their resource endowment (Section 2.2) for the establishment of cooperation structures. Concerning the state of empirical research, we focus primarily on studies about migrant organizations in Germany, since the societal, political and social context plays a major role in the foundation and further development of migrant organizations (Mualem Sultan 2022, p. 12f). For instance, political authorities and administration do not assign the same functions to migrant organizations in every country. At the same time, it has been stressed in international research that political expectations also play a role, for example, in how MOs are funded and promoted by politics and administration, or whether umbrella associations tend to be organized heterogeneously or homogeneously by origin (Odmalm 2004; Sardinha 2010; Frödin et al. 2021). In this context, when thinking about migrant organizations (in Germany as well as in other countries), the following question is of particular interest: to what extent may political–institutional opportunity structures (see, e.g., Fauser 2016, p. 28ff) not only determine the (political) success of an MO, but also influence (in a more fundamental sense) its specific goals and concrete operation mode (see e.g., Danese 2001; Amelina and Faist 2008; Halm 2013, p. 15); which may also include MOs' cooperation structures.

However, as far as our specific research question is concerned, the empirical data situation from other countries is rather patchy so far. Many international studies do not focus on MOs' cooperation patterns, but on their impact on integration in the host society and investigate only specific MOs (e.g., of a certain country/region). In the following Sections 2.1 and 2.2, we will therefore focus on empirical findings about the cooperation patterns of MOs in Germany. Where appropriate, we also refer to research findings from other countries that deal with the collaborative structures of MOs. First, however, it is important to point out some more general limitations in the state of research on MOs and the challenges behind them.

As mentioned above, in Germany, there is still relatively little research—especially when it comes to cross-regional or nationwide quantitative surveys—on the cooperation structures in the MO landscape. There are some very practical reasons for this: collecting (quantitative) data on MOs in Germany is difficult, as there is no complete register of these organizations. Thus, collecting MOs and their contact data requires much effort. Furthermore, conducting research without having personal contacts in migrant organizations and relevant stakeholders in the field reduces the probability of successful data collection, as the time resources of MO representatives are limited and the readiness of MOs to participate in a research project is related to personal trust. Hence, qualitative and explorative studies with a strong practical orientation dominate research on MOs in Germany. Another research gap is in the analysis of complex correlations and structures between different variables in quantitative research. These explorative studies are invaluable for the identification of relevant research questions, to gain insights into specific kinds of MOs and their challenges and to promote theory formation. To improve knowledge about MOs, we also need more research with a wider focus to give an overview of the landscape and to enable a comparative classification of qualitative results. Finally, although most studies address the question of MOs' cooperation and networking, only a few projects focus on this topic.

It is necessary to differentiate between publications (1) that analyze the conditions of the development of MOs, and those (2) that focus on the conditions for successful cooperation. In the following overview, we concentrate more on the former, as the aim of

this article is to analyze conditions for the probability of MOs to establish collaborations with other actors within and outside the MO landscape.

*2.1. The Relationship of Migrant Organizations' Activity Focus and Their Cooperation Patterns against the Background of Social Capital Theory*

To describe the relationship between MOs' fields of activity and their tendency to cooperate with other societal actors, we refer to the assumptions of the social capital theory of Robert D. Putnam (1995, 2020). For Putnam, social capital means "features of social life—networks, norms, and trust—that enable participants to act together more effectively to pursue shared objectives" (Putnam 1995, p. 664f). Following this basic definition, many researchers differentiate between social networks as structural social capital, and interpersonal trust and norms of reciprocity as cultural social capital (Freitag and Traunmüller 2008, p. 225f; Kunz et al. 2008, p. 42). Theoretical considerations and empirical findings concerning the relationship between these three aspects of social capital are not clear (Kriesi 2007, p. 27; Freitag and Traunmüller 2008, p. 226; Kunz et al. 2008, p. 46f). However, there is some support for a weak positive correlation between social networks and generalized interpersonal trust (Halman and Pettersson 2003, p. 176; Freitag and Traunmüller 2008, p. 230ff).

The question, especially for the relationship between generalized interpersonal trust and social networks, is important for Putnam's central distinction between two forms of social capital: bonding and bridging social capital. Bonding social or exclusive capital reinforces the relationships of (homogeneous) groups, while bridging or inclusive social capital focuses on the relationships and interactions between people who do not belong to the in-group (Putnam 2020, p. 22). Putnam cites inter alia ethnic fraternal organizations as an example of an organization that would typically produce bonding social capital, while the civil rights movement, for instance, functions as an example of a bridging social capital producing organization (Putnam 2020, p. 22). Thus, Putnam hypothesizes that bonding social capital only serves to strengthen trust between in-group members (e.g., members of a certain organization), while bridging social capital supports a more generalized social trust and a wider identity, which also includes out-group members (Putnam 2020, p. 23). Some scholars describe the difference between bonding and bridging social capital with reference to strong and weak ties (Vermeulen and Keskiner 2017). In this respect, the advantage of strong ties lies in the in-group solidarity and understanding, while weak ties offer the chance to get access to information and resources, which are not sufficiently available in the in-group (Vermeulen and Keskiner 2017, p. 303).

In the scientific public, the distinction between these two kinds of social capital has often been discussed with the tendency to divide organizations in good ones (those which produce bridging social capital) and problematic ones (those which produce bonding social capital). This focus overlooks the crucial point that bonding and bridging are just two traits of an organization, and that most civil organizations enhance both bonding and bridging social capital (Klie 2022, p. 130). Even Putnam points out that "bonding and bridging are not 'either or' categories in which social networks can be neatly divided, but 'more or less' dimensions along which we can compare different forms of social capital" (Putnam 2020, p. 23).[8] Looking at MOs that fight against discrimination, for example, we find both kinds of social capital are most likely to be combined. These MOs create a safe space for discriminated people and develop a strong in-group identity to empower their members not to accept unreasonable disadvantages. In this respect, the production of bonding social capital is an important process for the organization and their members. Many of these organizations are also politically active. Thus, in addition to creating a strong in-group identity, these groups approach the political class, as well as the whole of society, through the formulation of political demands, organizing demonstrations or antidiscrimination workshops. This example leads us to the second critical point against the positive/negative dichotomy of social capital, because even bonding social capital can provide important support for in-group members, which might eventually have a positive impact on wider

society. Putnam also mentions that both bonding and bridging social capital, under certain circumstances, might have positive effects (Putnam 2020, p. 23). The German sociologist Georg Elwert argued already in the early 1980s that immigrant networks can strengthen the self-confidence of migrants, convey knowledge about the receiving society and represent their interests, which supports the integration processes (Elwert 1982, p. 721ff).[9]

Based on these general theoretical arguments, we assume that weak ties that can be provided by bridging social capital are particularly important for the question of whether MOs enter into collaborations. Although we expect most MOs to produce both kinds of social capital[10], there might be differences between MOs depending on their (most important) activities. Two considerations play a crucial role here: first, to what extent MOs depend on cooperation with other actors to achieve their goals. Second, the question of cooperation also plays a role in the ascribed legitimacy of an organization. With regard to the second point, a distinction must be made between religious and secular organizations. Some scholars argue that religious MOs are legitimized primarily by the acceptance of their followers (cognitive legitimacy), while secular MOs can more easily legitimize themselves through the recognition and support of actors outside their own organization (sociopolitical legitimacy) (Vermeulen and Brünger 2014, p. 983). Although this assumption is based on the type of an organization, we also believe it to be suitable for the legitimacy of the MOs' main activities. Thus, we postulate that networks and cooperation are less important for cultural and religion-related activities, except with regard to cooperation with other religious organizations.[11] In contrast, we expect both political and social active MOs, to use cooperation with external actors to increase their legitimacy, as well as to reach their own goals. For political activities, however, we assume that they foster both internal ties and external relationships, as explained above. Hence, we expect them to maintain collaborations to reach their goals, especially with political actors. For social work, we believe that bridging social capital is also a crucial aspect to reach goals. Providing knowledge about Germany, as well as teaching the language, helping migrants to find a job and offering care are all activities, which help to improve participation of migrants in a host society and focus on life outside of the organization. More specifically, we assume that many of these offers and services can only be provided if MOs that offer social work, share their knowledge and expertise with other actors in the society. Contact and cooperation with the public institutions is necessary, especially for services involving legal aspects of life (e.g., demand of transfer payments). Therefore, we expect social work to be related to cooperation in particular. We assume that this is also true if the MOs have had negative experiences related to collaborations, as described in the introduction. Since they rely on cooperation with others, we rather suspect that in this case they actively seek such collaborations that they can actively shape.

Overall, regarding the German MO landscape, there is little empirical research on the impact of MOs' activities on their tendency to develop collaborations. The few studies that exist rather support our theoretical assumptions: Halm et al. (2020) expect MOs with a focus on social work to have a wider network than other kinds of MOs (p. 18), and find positive correlations between the support of refugees and the number of local cooperation partners—excluding local political actors (Halm et al. 2020, p. 90). In contrast to our theoretical assumptions, Klie (2022) reports, in her comparative study of secular and religious MOs in the North Rhine-Westphalia, that both kinds of MOs do not differ in the general probability of cooperation (p. 389). However, secular MOs cooperate more with public institutions, as well as other civil organizations, and have a wider network overall, while religious MOs primarily tend to cooperate with other religious organizations and have a smaller network overall (Klie 2022, p. 190ff).

*2.2. The Impact of Ressources and Structural Aspects on Collaborations*

Overall, empirical studies on migrant organizations in Germany focus more on the structural aspects that influence MOs' cooperation structures, like human and financial resources, than their activities. Hence, studies on the German MO landscape show a wide

network of cooperation partners among MOs, sometimes even more than non-migrant organizations (Priemer et al. 2017, p. 44; Klie 2022; SVR-Forschungsbereich 2020a). In contrast to the research on the influence of activity fields, there are hardly any systematic theoretical considerations to be found in research on the relationship between structural characteristics of MOs and their propensity to cooperate with other actors within and outside the MO landscape.

The results of a study by Huth (2011) show that most MOs in Germany have relatively few resources (p. 18).[12] In the context of this lack of resources, she concludes that MOs have in particular a need of a functional network with other (non-migrant) actors in civil society and public institutions (p. 39). Hence, we would expect a negative correlation between resources of MOs and cooperation with civil organizations.

However, one could also argue the opposite hypothesis, since collaborations also consume an organization's resources. In the analysis of welfare work in MOs by Halm et al. (2020), established MOs, especially those of the 'Gastarbeiter' generation, are more likely to cooperate with welfare associations than other MOs (p. 96f). Furthermore, the number of paid employees and unpaid volunteers correlate positively with the number of cooperation partners (Halm et al. 2020, p. 90). Klie (2022) also reports that the number of members and paid employees positively correlates with the number of cooperation partners (p. 293).

The qualitative study by Hunger and Metzger (2011), carried out on behalf of the Federal Agency of Migration and Refugees, analyzes the conditions for successful cooperation of MOs in Berlin with other actors. They differentiate between relational, external and internal factors. For our own research aim, their results relating to the internal factors are most relevant. While the structure of members and their target group does not have a strong impact on the success of a cooperation, fewer resources regarding volunteers and paid employees, a lack of rooms and less knowledge of process sequences (e.g., for funding) inhibit successful cooperation (Hunger and Metzger 2011, p. 68ff).

Halm and Sauer (2015) found, in their analysis of welfare service in Islamic religious organizations, that at least one-third cooperates with certain public local institutions and those of the civil society (p. 81). Overall, the number of paid employees and, to a lesser extent, the number of volunteers, correlate positively with cooperation with different public institutions and organizations in civil society in the sphere of youth and senior citizen work (Halm and Sauer 2015, p. 82f). Nevertheless, cooperation, especially with public institutions, is not only a matter of the number of paid employees or volunteers. The results from Halm and Sauer suggest that the number of qualified volunteers increases the probability of cooperation with public institutions in the social sector as well (Halm and Sauer 2015, p. 83; 2018, p. 324).

However, there is also empirical support for Huth's assumption. In the qualitative study by Barglowski and Bonfert (2022) less professionalized MOs have a wider network than more professionalized MOs, especially with other social/non-profit organizations (p. 9). They also conclude that cooperation with other social/non-profit organizations is particularly relevant for the work carried out by smaller, less professionalized MOs (p. 9). The quantitative analysis of 319 network partners in the German cities of Bochum, Dortmund and Duisburg by Günzel et al. (2022) confirms these results in a certain way. They show that state institutions (without explicitly political actors), welfare associations and other MOs are the most frequent cooperation partners of MOs that provide social security services (Günzel et al. 2022, p. 8). Furthermore, a high relevance of networks is primarily related to cooperation with government institutions and welfare organizations. Many members and paid employees, on the other hand, have a negative influence on the relevance of network relationships, even though the effect sizes are much lower here (Günzel et al. 2022, p. 9f). Nevertheless, overall results that demonstrate a positive correlation between human resources and the formation of collaborations predominate.

However, there definitely seem to be clear links between resources and cooperation structures. These results raise the question of whether MOs with a focus on social work have greater or fewer resources than other MOs. The resource mobilization theory assumes

that external support of organizations may channel the activities and action of a certain organization (Danese 2001, p. 72; Hein 1997, p. 283).[13] Assuming that both governmental and non-governmental institutions have an interest in improving the social participation of people with a migration background, it can be assumed that MOs with activities in social work, as well as political advocacy groups, are more likely to benefit from funding and, hence, have more human resources.[14]

Fauser (2016) states that MOs with public funding have a higher chance of offering a broad range of services, which might also have a positive effect on membership recruitment (p. 96 f). The study by the Expert Council on Integration and Migration shows that MOs, which are active in social work or political lobbying, partly have more paid employees, and have a higher probability of funding compared with MOs active in cultural maintenance or religion (SVR-Forschungsbereich 2020a, pp. 48f, 74f). Furthermore, Halm et al. (2020) report a weak positive correlation between the number of members and the number of different social activities, and a stronger correlation with membership in an association (p. 87f). Halm and Sauer (2023) report a positive correlation between MOs that focus on integration in Germany and are active in political advocacy and public funding (p. 61f). Frödin et al. (2021) report—in their study of the members of 52 MO umbrella associations in Sweden—a positive relationship between MO's size and the diversity of activity areas (p. 11).

*2.3. Hypotheses*

Based on the theoretical considerations and empirical findings described above, we can now formulate our research hypotheses. Regarding the correlation of personnel and financial resources and the fields of MOs' activities, we assume—in accordance with the previous results referred to—a positive correlation between social and political work and resources:

**H1.** *MOs, which are active in social or political work, have more resources in terms of volunteers, paid staff, and funding than MOs, which offer cultural maintenance and religious activities.*[15]

In accordance with the theoretical consideration that MOs with activities in social work and political advocacy have a higher need for cooperation to reach their goals and legitimize themselves, and in the context of previous studies that show a positive correlation between these fields of engagement and a wide network of cooperation partners, we assume:

**H2.** *If MOs offer social work, they are more likely to be member in an association and more likely to cooperate with public institutions and organizations of civil society. MOs with political activities are more likely to be member of an association and to cooperate with public institutions. Culture maintenance and religion correlate positively with cooperation with religious communities.*

**H2.1.** *Social work is strongly related to a high diversity of cooperation partners, political activities correlate moderately with the number of cooperation partners.*

Looking at existing research, the effects of personnel resources are less clear. On the one hand, there seems to be a positive effect on the number of cooperation partners when MOs can draw on a broad base of volunteers and paid employees. This is understandable, since building networks ties up human resources. On the other hand, some results indicate that small MOs with few personnel resources in particular have a special need for low-threshold cooperation with other associations. Therefore, a differentiation seems to be necessary, which takes into account the fact that paid staff is particularly important for cooperation with public institutions, as well as for membership in an association.

**H3.** *MOs with paid staff and many volunteers are more likely to be member of an association and more likely to cooperate with public institutions.*

**H3.1.** *With increasing number of paid employees and volunteers, the number of different cooperation partners increases.*

According to Hypothesis 3 and 3.1, we also believe that funding is more important for cooperation with public institutions.

**H4.** *MOs with funding are more likely to cooperate with public institutions.*

Next to these assumptions regarding certain aspects of MOs' resources, we additionally believe, in accordance with the results of Halm et al. (2020), that MOs that can be regarded as more established are also more likely to form an extensive network.

**H5.** *More established MOs are more likely to be a member of an association and to cooperate with other civil society organizations as well as with public institutions and have a higher number of different cooperation partners.*

### 3. Methods and Data

For the analyses of MOs' network and cooperation structure, we used the data from a standardized quantitative online-survey[16], conducted by the Research Unit of the Expert Council of German Foundations on Integration and Migration between January 2020 and March 2020 in the German federal states of Bavaria, Berlin, North Rhine-Westphalia and Saxony. The survey was part of the research project "Immigrant Organizations—Political and Civil Society Partners", which was funded by the Federal Ministry of the Interior, Building and Community from October 2018 to December 2020. The federal states were chosen based on the principle of most difference (Levy and Lemeshow 2008). Here, the federal states should differ as much as possible regarding the conditions that might affect the number and activities of the MOs. The criteria used to select the federal states were economical aspects, as well as the size of the population with a migration background. In addition, the integration policies in the states were considered. Regional criteria also played a role, as at least one eastern German state and one city state needed to be represented. Finally, only populous states were selected in order to generate a sufficiently large sample to allow quantitative analyses (SVR-Forschungsbereich 2020b, p. 5ff). Overall, 764 MOs participated in the survey: 132 from Bavaria, 109 from Berlin, 422 from North Rhine-Westphalia, 56 from Saxony, 44 from other federal states and one MO without information about their location (location of the headquarters)[17]. To establish an address basis for the survey, the research team set up a data bank of 6851 MOs[18] in these four federal states from April to December 2019. To reach almost all MOs that fit to the definition (see Introduction), the research team identified and contacted the relevant stakeholders of public institutions, as well as in organized civil society (e.g., MO roof associations) and asked for the names and addresses of known MOs. As the population of MOs in Germany is unknown, no statements concerning the representativeness of the sample are possible. For an impression of the sample quality, we compared the realized sample with the data bank. MOs from Bavaria are slightly underrepresented, while those from Berlin and Saxony and those located in big cities are slightly overrepresented (SVR-Forschungsbereich 2020a, p. 17). A further comparison of surveyed MOs with the data bank was not possible, because information about other variables like fields of activity, year of foundation, budget or number of employees were only available for few organizations in the data bank.[19]

To measure cooperation, we use the following variable: "Who do you work with regularly?" MOs had the chance to mention several partners out of 15 options.[20] For our analysis, we re-categorized the original variable. We classified the answers "state policy/administration (e.g., state ministries)"; "federal policy/administration (e.g., federal ministries)"; "municipal administrations"; and "district administrations" as "*cooperation with public institutions*" and the answers "other migrant organizations"; "welfare associations"; and "other non-profit associations/organisations" as "*cooperation with organizations of civil society*". Cooperation with churches and religious communities were kept as a separate option, because the results discussed above suggest a difference in cooperation structures between religious and other civil society organizations, especially for MOs with religious offerings. We did not take into account the other cooperation partners, as the

descriptive analyses show that they are only of minor importance for MOs' cooperation patterns (SVR-Forschungsbereich 2020a, p. 59).[21] As a specific form of network, we took into account whether or not an MO is member of a roof association. For our analysis, we combined MOs with one membership and those with several memberships to one category. Additionally, we formed a variable to measure the diversity of cooperation partners. Therefore, we counted the number of different cooperation partners the respondents mentioned based on the original question. Hence, this variable varies between 0 and 15.

To measure activities, we asked MOs which areas they made offers most frequently in 2019. Respondents could mention up to five[22] out of 23 options.[23] We also re-categorized this variable with respect to three broader areas of activity: (1) social work and protection; (2) political activities and lobbying; and (3) cultural maintenance and religion. Social work is measured through engagement in the areas "employment agency"; "consulting"; "education"; "health"; and "support for refugees". With this classification, our aim was not to define social work and protection with too much restriction, but also not to include spheres where some engagement might belong to social work, but other activities do not (e.g., child, youth and senior citizen work, specific offerings for women). Political activities and lobbying include "anti-discrimination work"; "political representation of interests"; and "representation of professional interests". Finally, "artistic-cultural activities (e.g., music, theatre)"; "cultivation of culture(s) of origin"; "cultivation/teaching of language of origin"; and "religion" were classified as cultural and religious activities. Thus, in this paper, we do not differentiate different *types of MOs* but different *types of activities*, which we regard as traits of MOs. Consequently, MOs are not social work organizations or political organizations or cultural and religious MOs. Some of them might be active in all three areas, some in none of them. We chose this approach because most MOs are multifunctional, and a classification of MO types will not reflect their wideness of engagement accurately.[24]

For the measurement of MOs' human resources, we used the number of volunteers and the number of paid employees.[25] As an indicator of material resources, we take into account whether an organization gets funding and/or donations.[26] Finally, to get an impression of how established an MO is, we added two more variables as predictors: first, the question of whether or not the surveyed organization has its own management; and second, the year of foundation. Together with human and financial resources, we assume this provides an appropriate measurement of how established a certain MO is.

To test whether or not offers of social work and political activities are accompanied by more human and material resources than religious and cultural activities (H1), we analyzed the relationship between areas of engagement and resources with descriptive methods. The other hypotheses were tested based on multiple logistic and linear regression models. We calculated four logistic regression models with dependent variables: (1) membership in a roof association; (2) cooperation with public institutions; (3) cooperation with organizations of civil society; and (4) cooperation with religious communities. For correlations with the diversity of cooperation partners, we calculated a linear regression model (OLS) with the number of different cooperation partners as a dependent variable. As predictors, we used the categorical variables fields of activity (social work, political work, cultural and religious activities), funding, existence of management, the metric variable number of volunteers and paid employees and year of foundation.[27] For funding[28], we combined MOs with funding and with combined funding and donation into one category. Next, we divided the metric variables into four categories based on quartiles, with the lowest category always as the reference. Then we divided the metric variables into four categories based on quartiles, with the lowest category always as the reference. We did this because the distribution of these two variables is skewed due to there being few MOs with a high number of paid employees and volunteers, which might affect the estimation of coefficients.[29]

As our analysis is based on cross-sectional data, it is only possible to test correlations, but not to prove the direction of a certain correlation. Although we assume that activities and resources theoretically influence the probability of networking and cooperation by the

model specification, we cannot exclude the possibility that the reverse is the case. Thus, it is important to consider this while interpreting the empirical findings.

## 4. Results

### 4.1. Descriptive Results

Before testing the formulated hypotheses, we will briefly look at the new categorized activity areas. The results confirm that, from a quantitative perspective, social work is the most important of the three fields of activity studied here (Table 1). More than three quarters of the MOs studied are engaged in this field. However, there are, again, clear focal points within social work: education (42%); support for refugees (36%); and consulting (33%) (SVR-Forschungsbereich 2020a, p. 23). In contrast, health and employment services are much less represented, at 8% and 5%, respectively (SVR-Forschungsbereich 2020a, p. 23). Almost 60% have a focus on cultural maintenance and religion. It should be noted, however, that artistic activities are indicated more frequently (39%) than maintenance of the culture of origin (19%) or religion-related activities (15%). In contrast, looking at the quantitative share, political activities and lobbying are less typical for MOs. Almost 30% are politically active. This finding fits with the fact that local organizations/associations typically delegate political activities (especially in the area of political advocacy) in part to umbrella associations/federations on the federal state or even national level. Finally, we will briefly discuss the correlations between the three activity areas. Social and political work tend to correlate slightly positively (Cramer's V = 0.08*), while both correlate negatively with cultural and religious activities.[30]

**Table 1.** Description of the fields of activities.

| Fields of Activities | % | Number of Volunteers (Mean) | Number of Employees (Mean) |
|---|---|---|---|
| Social work | 77 | 42.0 | 6.7 |
| Political activities | 29 | 56.8 | 7.4 |
| Culture and religion | 59 | 42.9 | 3.6 |

Source: SVR-Forschungsbereich (2020a, 2020b).

To test whether social work and political engagement are associated with more human resources and more funding frequency compared to cultural cultivation and religion (H1), we first take a comparative look at the average number of volunteers, as well as paid employees (Table 1).[31] The results show that social as well as political work is accompanied by more staff, with 6.7 and 7.4 paid employees, respectively, than is the case for cultural and religion-related activities, which have an average of only 3.6 paid employees. H1 can therefore be confirmed for the number of paid employees. In terms of the number of volunteers, however, only political activities stand out, with an average of 56.8 volunteers, while the other two areas of engagement have almost the same number of volunteers: 42 and 43, respectively.[32] We cannot confirm H1 for the number of volunteers in social work, but can for the equivalent in political activities. Furthermore, the means of both variables tend to increase because of there being few MOs with many volunteers and/or employees. The median for the number of volunteers in all of our three activity fields is 20. Hence, the difference between political engagement and the other activity fields is related to the few MOs with a huge number of volunteers. The median of paid employees is also much lower. Half of MOs with a focus on social work and culture and religion-related activities employ one person, while half of those doing political work have three employees.

H1 is also confirmed when it comes to the share of MOs which get funding (Figure 1). While 55% of MOs with social work as focal point and 59% of politically active MOs get both funding and donation, this share is lower for MOs specializing in cultural and religious activities with just 47%.[33] The share of cultural and religious active MOs that only get funding (without donation) is 16%, which is also slightly lower than that of MOs offering social (19%) and political work (21%). Only the share of organizations that profit

from donation is higher for MOs with cultural/religious activities than for the other two activity fields. This can probably be explained by the fact that donation is particularly important for the work of religious migrant organizations (Ceylan and Kiefer 2016, p. 85). If we look to the number of MOs that receive funding or donations separately, a strong difference between activity fields only exists for funding, while the difference for those that only receive donations is comparatively lower. While 80% of politically active MOs get funding, only 63% of MOs with an emphasis on culture and religion do. MOs that offer social work are in between with 73%. The differences regarding donation are much lower with values of 66% (culture and religion); 68% (social work); and 69% (political activities). Overall, H1 can be confirmed. However, a difference between political and social activities was found, in that the former have more resources overall than the latter.

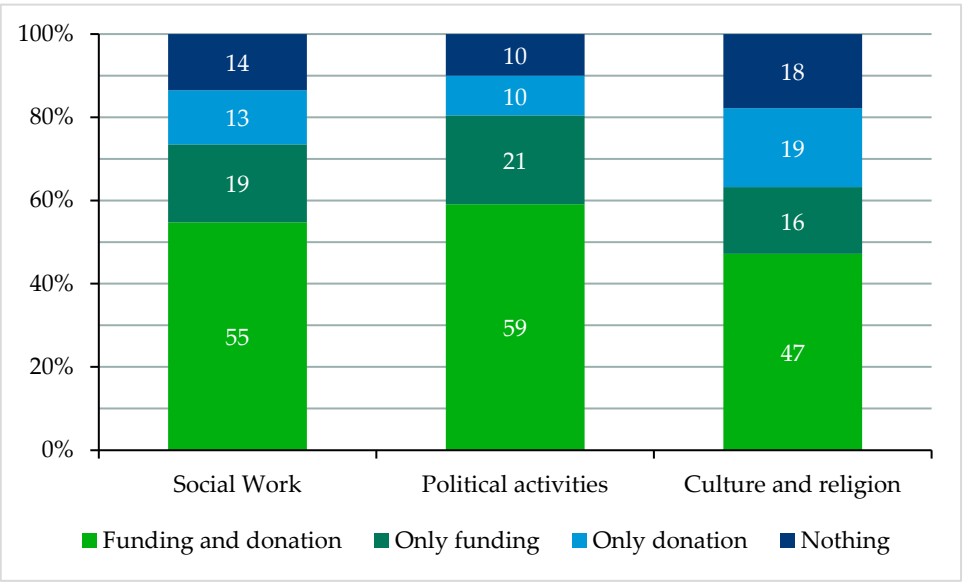

**Figure 1.** Funding by fields of activities.

### 4.2. Multivariate Results

To test hypotheses H2 to H5, we calculated multiple logistic and linear regression models. Table 2 shows the results of the logistic regressions with the dependent variables being membership in an association and cooperation with (1) public institutions; (2) civil society; and (3) religious organizations.[34] The Pseudo-$R^2$ of the four regression models varies between 8% and 12%, suggesting that there are probably other factors besides the variables tested here that influence the likelihood of network and cooperation formation.

In contrast to our assumption, MOs with paid employees differ regarding their probability to have cooperation partners in religious communities (H3), MOs with one paid employee are more likely to cooperate with religious organizations than completely volunteer-based MOs. However, there is no effect when applied to two or more employees. Thus, we need to reject H3 with respect to cooperation with religious organizations. Nevertheless, the probability of cooperation with civil society and religious communities seems—with respect to our data—to be relatively independent of the staffing level. Contrarily, H3 is true for the probability of being a member of an association and of cooperating with public institutions. MOs with two or more employees are more likely to hold a membership in an association and, in this respect, professionalized MOs with more than five employees are more likely to cooperate with public institutions. For cooperation with public agencies in particular, these results deliver indications that MOs must already be relatively highly professionalized. This can possibly be interpreted as an indication that public agencies prefer established MOs as cooperation partners (H5). With regard to the number of volunteers, H3 needs to be completely rejected. The number of volunteers correlates positively with the likelihood of collaborations with civil society organizations and religious communities,

especially if MOs can rely on the support of many volunteers. In summary, we can say that the number of paid employees is primarily relevant for association networks and cooperation with public agencies, while, within civil society, volunteerism is decisive.

**Table 2.** Factors influencing the probability of being a member in an association and of cooperation (logistic regressions).

| Predictors | Membership in Association | Cooperation with . . . | | |
| --- | --- | --- | --- | --- |
| | | Public Institutions | Civil Society | Religious Communities |
| Social work: yes | 1.40 | 1.78 ** | 3.33 *** | 1.33 |
| Political activities: yes | 1.79 * | 2.01 ** | 1.55 | 0.82 |
| Cultural and religion: yes | 0.77 | 0.93 | 0.97 | 1.02 |
| 10 to 20 volunteers [a] | 0.93 | 1.38 | 1.24 | 1.95 ** |
| 21 to 40 volunteers [a] | 1.23 | 1.28 | 1.74 | 1.70 |
| 41 and more volunteers [a] | 1.45 | 1.15 | 2.45 * | 3.51 *** |
| 1 employee [b] | 1.41 | 1.08 | 1.19 | 2.19 * |
| 2 to 5 employees [b] | 2.13 ** | 0.61 | 1.42 | 1.04 |
| 6 and more employees [b] | 1.99 * | 2.60 ** | 1.36 | 1.07 |
| Existence of management: yes | 1.32 | 1.26 | 1.28 | 1.09 |
| Funding: yes | 1.48 | 2.05 ** | 1.89 * | 0.41 *** |
| Year of Foundation: 1992 to 2004 [c] | 0.58 * | 1.25 | 1.51 | 0.82 |
| Year of Foundation: 2005 to 2013 [c] | 0.49 * | 0.57 * | 1.80 | 0.66 |
| Year of Foundation: after 2013 [c] | 0.28 *** | 0.60 | 1.81 | 0.73 |
| Pseudo R$^2$ | 0.11 *** | 0.11 *** | 0.12 *** | 0.08 *** |
| Number of cases | 591 | 606 | 606 | 606 |

[a] reference: less than 10 volunteers; [b] reference: no paid employees; [c] reference: before 1992; coefficient: odds-ratio; significance: *** $p < 0.001$; ** $p < 0.01$; * $p < 0.05$. Source: SVR-Forschungsbereich (2020a, 2020b).

It is correct that MOs with funding are more likely to cooperate with public institutions (H4). However, funding also increases the probability of cooperation with associations of civil society, and has a slightly negative effect on the probability of cooperation with religious communities. However, looking at the effect of funding on cooperation with public institutions, the problem arises that MOs may have understood public funding as a form of cooperation (Hunger and Metzger 2011). If a model controls only for funding from private donors, there is no significant effect.

Regarding the correlations with the activity areas: overall, we observe that MOs active in social and in political work have a tendency to be more likely to cooperate than MOs that are not active in these fields. Therefore, H2 is largely correct. Nevertheless, there are fewer significant effects than expected. Activities in the field of social work correlate positively with cooperation with public institutions and organizations of civil society, but not with a higher probability of being a member of an association. This is surprising, as welfare organizations (Wohlfahrtsverbände) play an important role in the organization of offers for social work in Germany. Furthermore, almost one-quarter of MOs that are members of an association, are a member of such a welfare organization (SVR-Forschungsbereich 2020a, p. 54). It seems that many MOs organize their support offers independently of the established welfare structure in Germany. If we recalculate the model with membership in an MO association, we observe a positive effect of social work. The fact that we do not find a significant correlation with general association membership is apparently because MOs in social work seem to be less likely to be active in non-migrant associations. As expected, politically active MOs are more likely to be a member of an association and to cooperate with public institutions. As expected, a focus on cultural maintenance and religion in MOs' work has no correlation with cooperation with state institutions and organizations of civil society and membership. Contrary to our assumption, there is no significant effect

for cooperation with other religious communities. This can probably be explained by the fact that this field of activity is more strongly influenced by culture than by religion. Furthermore, it seems that cooperation with organizations of civil society and networking via membership of an association are two clearly distinguishable forms of networking. If we include association membership as a predictor in the regression model of cooperation with civil society, membership has no significant effect.

To test if more established MOs are more likely to maintain cooperation and networks (H5), an overall view on the results is necessary, as paid employees and volunteers as well as successful acquisition of funding are also indicators of how established an organization already is. Additionally, we added the existence of a (professional) management and the foundation year as predictors for our analysis. Although we found some effects from the number of employees, the question of whether or not an organization has a (professional) management has no effect on cooperation and membership. The year of foundation almost only effects the probability of being member in an association. The younger an MO is, the less likely it is to be an association member. Although this result was expected in general, the strength is nevertheless surprising, because even MOs, which have been founded after 1991, are less likely to be member of an association. The only additional effect we found is that MOs founded between 2005 and 2013 are also less likely to collaborate with public institutions. That this effect does not exist for MOs founded after 2013 might be related to the fact that in this last period many (self)organizations of and for refugees have been founded, and funding programs focused more and more on the integration of refugees. With regard to the logistic regressions, H5 cannot be confirmed in its general formulation. We find expected effects for almost all predictors. However, the resources that are finally important for cooperation seems to be primarily related to the specific type of cooperation partner.

After the question of cooperation with specific partners, we also investigated the relationship between fields of activities and structural condition and cooperation with different partners on the basis of multivariate linear regression models (Table 3). In other words, we look closer at the diversity of cooperation of MOs.

**Table 3.** Factors influencing the diversity of cooperation partners (OLS regression).

| Predictors | Number of Different Cooperation Partners |
|---|---|
| Social work: yes | 0.18 *** |
| Political activities: yes | 0.14 *** |
| Religious and cultural activities: yes | −0.06 |
| 10 to 20 volunteers [a] | 0.11 ** |
| 21 to 40 voluntieers [a] | 0.09 * |
| 41 and more voluntieers [a] | 0.19 *** |
| 1 employee [b] | 0.09 * |
| 2 to 5 employees [b] | 0.02 |
| 6 and more employees [b] | 0.24 *** |
| Existence of management: yes | 0.10 * |
| Funding: yes | 0.10 ** |
| Year of Foundation: 1992 to 2004 [c] | 0.07 |
| Year of Foundation: 2005 to 2013 [c] | 0.01 |
| Year of Foundation: After 2013 [c] | 0.01 |
| Adj. $R^2$ | 0.25 *** |
| Number of cases | 606 |

[a] reference: less than 10 volunteers; [b] reference: no paid employees; [c] reference: before 1992; coefficient: standardized regression coefficient beta; significance: *** $p < 0.001$; ** $p < 0.01$; * $p < 0.05$. Source: SVR-Forschungsbereich (2020a, 2020b).

The model explains 25% of the variance (Adj. $R^2$), which is acceptable given the number of predictors. As expected, MOs with a focus on social and political work tend to have many different cooperation partners. However, the effect of political activities is slightly lower than that of social work. In contrast, cultural and religious activities have no effect on the number of cooperation partners.

According to our assumptions, paid and unpaid staff also correlate positively with diversity of cooperation, which is in agreement with the hypothesis that MOs need human resources for cooperation. While the number of different collaborations increases in general with the increase of volunteers, we only observe effects for MOs with one employee and with six or more employees. Although more research is needed for a more detailed proof, we postulate that there is one huge difference between completely volunteer-based MOs and MOs that have at least one paid employee. Even one employee makes it easier to maintain contacts and cooperation. It also is consistent with the fact that the existence of a (professional) management increases the number of different cooperation partners. Next to this result, we see that highly professional working MOs with a huge paid staff are able to maintain a wider network of cooperation partners. Finally, the positive effect of funding on cooperation with certain partners (presented in Table 2) exists in a linear regression model as well.[35]

In contrast to the logistic models, the linear model also allows an interpretation of the effect sizes, so that the effects of the individual predictors can be compared. Six or more employees have the strongest effect on the number of cooperation partners, followed by 41 or more volunteers. The effect of social work is almost as strong as the one of volunteers; the effect of political activities follows at some distance. Two conclusions can be drawn from this. First, human resources and content orientation of MOs are equally important for the diversity of cooperation partners. Second, the effects of human resources are stronger when the MOs are very well equipped. Developing and maintaining a broad network with many different partners, thus, obviously requires significant human resources. Overall, more established MOs have a wider network as expected (H5), although there is no effect with regard to year of foundation.

Beyond the direct effects reported here, it seems plausible to assume that the opportunities for cooperation in social work and political engagement increase when MOs, active in these areas, have extensive human resources at their disposal. To test this, we calculated further regression models to predict the diversity of cooperation partners, including interactions between the number of paid employees and activities (Table A1, Models 1 to 3), and the number of volunteers and activities (Table A1, Models 4 to 6). We also did the same for a possible interaction between funding and the activity areas (Table A1, Models 7 to 9). Here, too, the question arises as to whether funding is of different importance for building collaborations, depending on the content orientation of an MO. Since there are only very few interaction effects, we do not discuss these results in detail. Nevertheless, we would like to briefly mention the few significant effects at the end of this section. First, there were no interaction effects for the number of volunteers and the activity fields. Furthermore, there were also no effects for the interaction between the number of paid staff and social work. Thus, MOs with a focus on social work tend to cooperation, independently of their human resources. This underlines our assumption that cooperation is a necessary precondition for implementing activities in social work. For political work, we notice that this activity is accompanied with more cooperation partners, if there is one paid employee. The assumption that even one paid employee facilitates the establishment of collaborations thus seems to apply primarily to MOs that are politically active.[36] For the interaction between funding and activities, we can neither report effects for social work nor for culture and religion. There is only one effect between political activities and funding to explain the number of different cooperation partners. Politically active MOs, which do *not* get funding, have significantly more cooperation partners than MOs with other activities. At first glance, this result might be surprising. Our interpretation is that political engagement is accompanied by a wide cooperation network, even if there is no funding. This can be

explained by the necessity for these MOs to address the public anyway. However, for MOs with other content foci, funding increases the likelihood of cooperation.

Summarizing the results, MOs that offer social work or are politically active have more resources at their disposal than MOs that are active in the area of culture and religion. However, this is even more true for political engagement. Furthermore, it can be determined that both the resources and activities of an MO are important for the propensity to cooperate, although the correlations differ depending on the cooperation partner. In addition, human resources are more important overall than whether or not an organization receives funding. When it comes to cooperation with other civil society organizations, the number of honorary employees is the most important consideration, while the number of paid employees is decisive for cooperation with public agencies and for association membership. Furthermore, it seems that a relatively broad base of volunteers and paid employees is necessary to establish and maintain cooperation (Naumann 2011, p. 48). In addition, social work and political commitment are positively related to cooperation overall. While MOs that offer social work are more likely to cooperate with public institutions, politically active MOs are also more likely to organize themselves in an association, while MOs that offer social work are more likely to cooperate with other civil society organizations.

## 5. Discussion

The analyses have confirmed that personnel resources in particular, as well as the content-related orientation of the MOs, are related to the tendency toward cooperation and network formation. Thus, our findings can (partly) confirm the results of other studies on a broader data basis. In addition, we were able to show that the correlations vary depending on the cooperation partner. Therefore, future research should take a closer look at the relationships to specific cooperation partners than has been the case so far.

For MOs active in social work, it has been confirmed that cooperation with other actors seems to be central to their work. This is true regardless of whether they have paid or volunteer staff, as the more detailed analysis of interactions has shown. Moreover, these MOs seem to be less integrated into association structures outside the MO landscape than we expected. Further research of MOs that are not member of an association is needed.

At the same time, however, it is important to point out some limitations of the present study. We used data of a survey that was designed to provide an overview of the landscape of migrant organizations in Germany. Accordingly, it was not possible to go into as much depth in the individual topic areas as would have been desirable. For example, respondents were not asked about different concepts of cooperation (Hunger and Metzger 2011), but were able to base the survey on their own understanding. Thus, funding by public institutions seems to be understood as a type of cooperation. Furthermore, there might be huge differences in the frequency of contact with different partners (Reinecke et al. 2010, p. 70). Again, more detailed research on the type of cooperation is necessary. More detailed analysis would also be needed on the influence of volunteers and paid staff, taking into account, for example, how much time volunteers and paid staff invest per week or month. Furthermore, a closer look to activities and services MOs offer in the sphere of social work could help to establish a deeper understanding of the results. It seems that MOs more often provide informal and low-threshold social work services (Günzel et al. 2022, p. 7), which could also explain the results on membership in an umbrella association.

Another critical point concerns the question of the direction of the correlations. It has already been pointed out that cross-sectional data can, in any case, only test correlations, not a causal direction. Nevertheless, with the specification of the regression models, we assumed that fields of activity and resources influence the formation of networks and collaborations. At the same time, results from qualitative studies show that we are dealing with a complex web of interrelationships. Thus, in some places, a different direction of the interrelationships would also be conceivable. For example, it could also be the case that cooperation influences the content of an MO's work. This could be the case, for example, if MOs are approached by other organizations/institutions, and are won over as cooperation

partners in order to gain access to a specific (migrant) target group. A similar argument can be made with regard to the effect of funding, as the resource mobilization theory assumes (Hein 1997). Thus, the existence of a project partner is usually a prerequisite for tandem funding programs. In order to be able to examine the influences *on* cooperation and the influences *of* cooperation more precisely in terms of the direction of impact, it would be desirable to provide scientific support for such programs, which also have networking between MOs and other actors as a goal or as a component.

**Author Contributions:** Conceptualization, N.F. and M.M.; methodology, N.F.; software, N.F.; validation, N.F. and M.M.; formal analysis, N.F.; investigation, N.F. and M.M.; resources, N.F.; data curation, N.F.; writing—original draft preparation, N.F. and M.M.; writing—review and editing, N.F. and M.M.; visualization, N.F.; supervision, N.F. and M.M.; project administration, N.F. and M.M.; funding acquisition, Research Unit of the Expert Council on Integration and Migration. All authors have read and agreed to the published version of the manuscript.

**Funding:** This research is based on the results of a project carried out by the Research Unit of the Expert Council on Integration and Migration (SVR). It was written by the authors in their capacity as researchers within SVR's scientific staff. The project was funded by the German Federal Ministry of the Interior, Building and Community, grant number H I 2—41008/2#10.

**Institutional Review Board Statement:** According to both the ethics guidelines of the German Research Foundation (DFG) (https://www.dfg.de/en/research_funding/faq/faq_humanities_social_science/index.html, accessed on 2 April 2023) and the code of ethics of the German Sociological Association (DGS) (https://soziologie.de/dgs/ethik/ethik-kodex, accessed on 2 April 2023), approval by an ethics committee was not mandatory for the present study.

**Informed Consent Statement:** Informed consent was obtained from all subjects involved in the study.

**Data Availability Statement:** The data presented in this study are available on request from the corresponding author. A publication of the entire data set is planned.

**Acknowledgments:** We would like to thank the Expert Council on Integration and Migration gGmbH for the opportunity to use the data from the project "Immigrant Organizations—Political and Civil Society Partners" for this publication and the entire team for the constructive exchange. We thank Egemen Kepekçi for critically reviewing the manuscript. Finally, we would like to thank all the organizations that participated in the survey and thus made the study possible.

**Conflicts of Interest:** The authors declare no conflict of interest. The funders had no role in the design of the study; in the collection, analyses, or interpretation of data; in the writing of the manuscript; or in the decision to publish the results.

## Appendix A

**Table A1.** Factors influencing the diversity of cooperation partners with interaction effects (OLS regressions).

| Predictors | Number of Different Cooperation Partners | | | | | | | | |
|---|---|---|---|---|---|---|---|---|---|
| | M 1 | M 2 | M 3 | M 4 | M 5 | M 6 | M 7 | M 8 | M 9 |
| SW [a]: yes | 0.19 * | 0.18 * | 0.17 * | 0.16 * | 0.19 * | 0.18 * | 0.19 * | 0.18 * | 0.18 * |
| PA [b]: yes | 0.14 * | 0.08 | 0.13 * | 0.14 * | 0.12 | 0.14 * | 0.14 * | 0.36 * | 0.14 * |
| RCA [c]: yes | −0.06 | −0.06 | 0.09 | −0.06 | −0.06 | −0.04 | −0.06 | −0.06 | −0.09 |
| 10 to 20 volunteers [d] | 0.11 * | 0.11 * | 0.10 * | 0.12 | 0.12 * | 0.11 | 0.11 * | 0.12 * | 0.11 * |
| 21 to 40 volunteers [d] | 0.09 * | 0.08 * | 0.08 * | 0.11 | 0.08 | 0.07 | 0.09 * | 0.09 * | 0.09 * |
| 41 and more volunteers [d] | 0.19 * | 0.19 * | 0.17 * | 0.09 | 0.17 * | 0.25 * | 0.19 * | 0.20 * | 0.19 * |
| 1 employee [e] | 0.09 | 0.04 | 0.25 * | 0.09 * | 0.09 * | 0.08 * | 0.09 * | 0.08 * | 0.09 * |
| 2 to 5 employees [e] | 0.08 | 0.00 | 0.10 | 0.02 | 0.02 | 0.02 | 0.02 | 0.02 | 0.02 |
| 6 and more employees [e] | 0.19 | 0.24 * | 0.39 * | 0.24 * | 0.24 * | 0.24 * | 0.24 * | 0.25 * | 0.24 * |
| Existence of management: yes | 0.10 * | 0.10 * | 0.10 * | 0.10 * | 0.10 * | 0.10 * | 0.10 * | 0.10 * | 0.10 * |
| Funding: yes | 0.10 * | 0.10 * | 0.11 * | 0.11 * | 0.10 * | 0.10 * | 0.11 | 0.16 * | 0.07 |

**Table A1.** *Cont.*

| Predictors | Number of Different Cooperation Partners | | | | | | | | |
|---|---|---|---|---|---|---|---|---|---|
| | **M 1** | **M 2** | **M 3** | **M 4** | **M 5** | **M 6** | **M 7** | **M 8** | **M 9** |
| Foundation: 1992 to 2004 [f] | 0.07 | 0.08 | 0.07 | 0.07 | 0.07 | 0.07 | 0.07 | 0.08 | 0.07 |
| Foundation: 2005 to 2013 [f] | 0.01 | 0.01 | 0.01 | 0.01 | 0.01 | 0.01 | 0.01 | 0.02 | 0.01 |
| Foundation: After 2013 [f] | 0.01 | 0.01 | 0.02 | 0.01 | 0.01 | 0.01 | 0.01 | 0.00 | 0.01 |
| SW*1 employee | 0.00 | | | | | | | | |
| SW*2 to 5 employees | −0.07 | | | | | | | | |
| SW 6 and more employees | 0.05 | | | | | | | | |
| PA*1 employee | | 0.11 * | | | | | | | |
| PA*2 to 5 employees | | 0.05 | | | | | | | |
| PA*6 and more employees | | 0.02 | | | | | | | |
| RCA*1 employee | | | −0.21 * | | | | | | |
| RCA*2 to 5 employees | | | −0.10 | | | | | | |
| RCA*6 and more employees | | | −0.21 * | | | | | | |
| SW*10 to 20 volunteers | | | | −0.02 | | | | | |
| SW*21 to 40 volunteers | | | | −0.03 | | | | | |
| SW*41 and more volunteers | | | | 0.11 | | | | | |
| PA*10 to 20 volunteers | | | | | −0.02 | | | | |
| PA*21 to 40 volunteers | | | | | 0.02 | | | | |
| PA*41 and more volunteers | | | | | 0.04 | | | | |
| RCA*10 to 20 volunteers | | | | | | 0.00 | | | |
| RCA*21 to 40 volunteers | | | | | | 0.02 | | | |
| RCA*41 and more volunteers | | | | | | −0.08 | | | |
| SW*funding | | | | | | | −0.01 | | |
| PA*funding | | | | | | | | −0.27 * | |
| RCA*funding | | | | | | | | | 0.04 |
| Adj R$^2$ | 0.25 * | 0.25 * | 0.27 * | 0.25 * | 0.25 * | 0.25 * | 0.25 * | 0.26 * | 0.25 * |
| Number of cases | 606 | 606 | 606 | 606 | 606 | 606 | 606 | 606 | 606 |

[a] Social work; [b] political activities; [c] religious and cultural activities; [d] reference: less than 10 volunteers; [e] reference: no paid employees; [f] reference: before 1992; coefficient: standardized regression coefficient beta; significance: * $p < 0.05$. Source: SVR-Forschungsbereich (2020a, 2020b).

## Notes

[1] This increased need for diversity-sensitive social work goes hand in hand with a general increase in social service offers in the non-profit sector of civil society (Köcher and Haumann 2018, p. 37; Priemer et al. 2019, p. 48).

[2] The landscape of migrant organizations in Germany is very diverse and developing dynamically (see also SVR-Forschungsbereich 2020a). For this reason, there is no single definition that fits all civil society, political and academic contexts equally, and on which all migrant organizations can agree with at the same time. The coexistence of different definitions also leads to terminological diversity. As a result, e.g., the terms migrant organization and migrant association are sometimes used synonymously in the scientific literature. However, in the pesent article, the term migrant association is only used with reference to umbrella associations of MOs or other alliances consisting of a majority of institutional members (predominantly migrant organizations), and which have (among other possible activites) a focus on the representation of interests towards third parties (e.g., other associations, political authorities or the public).

[3] The definition closely follows that of Pries (2010, p. 16), but emphasizes somewhat more strongly that migration- or integration-related issues must be decisive for the organizations' self-understanding. For a deeper discussion of our definition of MOs see (SVR-Forschungsbereich 2019, 2020a).

[4] Although social work of MOs is in many countries of a certain importance, international studies show some differences compared to the situation in Germany. For Spain, Fauser (2016) points out that MOs historically focus in particular on political activities (p. 100). Swedish MOs, in contrast, focus primarily on cultural activities (90%), although more than a half are also engaged in political and social work (Frödin et al. 2021, p. 11). Finally, a study of MOs in Luxembourg suggests that MOs activities change depending on the needs of their target group. While MOs of a sub-Saharan background primarily offer social services, MOs of ex-Yugoslav migrants transformed from social work to cultural maintenance organizations (Gerstnerova 2016, p. 424).

[5] The need for MO in the field of social work increased again after 2005, due to social and integration policy changes. On the one hand, the welfare state increasingly relied on the voluntary engagement of civil society organizations; while on the other

hand, the requirements for integration policy changed with the recognition of Germany as an immigration country (Latorre and Zitzelsberger 2011, p. 54; Kellmer et al. 2022, p. 401f).

6    This term refers to the diverse field of non-profit organizations which serve the general interest and which are neither connected to nor managed by state authorities.

7    The authors distinguish between cooperation with an umbrella association, coaching/mentoring, tandem-projects, project conception, publicity actions and access to a certain target group (Hunger and Metzger 2011, p. 11).

8    That MOs can fulfill different functions for their members has already been described by Breton (1964), who pointed out that an institutionalized innerethnic infrastructure (provided by MOs) might, on the one hand, substitute institutions of the host country and enhance the relevance of migrants' national identity to a certain degree. On the other hand, this infrastructue can also bring important issues and challenges of people with a migration background to the public (p. 198f).

9    Following Elwert's hypotheses of integration within the ethnic group a controversial discussion between different researchers came up, if migrant organizations enhance or reduce integration processes (the so-called Esser–Elwert controversy). As this article does not focus on MOs' effects on integration and furthermore rejects the 'either or' perspective in this context, we will not repeat the arguments of this discussion. However, the debate of the effects of bonding and bridging social capital in a dichotomous perspective suggests that the Esser–Elwert controversy continues under a new label (Klie 2022, p. 130).

10    Next to the fact that most studies show a wide network of cooperation partners of MOs, over 90% of MOs also open their offers to non-members (SVR-Forschungsbereich 2020a, p. 19), which supports the assumption that bridging social capital is an important aspect of MOs' work.

11    In this article, we use the bonding/bridging concept only on the organizational level, not on the individual level of MOs' members or clients.

12    In an explorative Study, Huth interviewed 85 MOs in Hesse at the end of 2011 (Huth 2011, p. 9). Due to the small number of cases, the results give more impressions, but cannot deliver statically reliable results.

13    In Belgium, for instance, the state forces to join umbrella associations to gain access to public funding (Hooghe 2005, p. 980).

14    Although resource mobilization theory bear primarily on financial support, there is empirical evidence for a strong correlation between financial support (e.g., funding) and the development of a staff (Fauser 2016, p. 94; SVR-Forschungsbereich 2020a, p. 75ff).

15    This hypothesis has already been tested based on the individual activities and three MO types developed by a cluster analysis. In this article, we re-categorized the activities into broader activity areas (see section Methods and Data), which makes it necessary to look again at this relationship.

16    To increase response rates, the invitation via e-mail was combined with contacting by telephone, if both e-mail address and telephone number were available. If there was only a telephone number, interviewers asked after the e-mail address within the call to send an invitation. We chose this procedure also to get a less selective sample.

17    Next to the personal invitations of MOs in our data bank, some MO associations also sent the link to their members via e-mail to invite MOs that are not located in the four federal states to participate as well.

18    4.122 in North Rhine-Westphalia, 1.766 in Bavaria. 732 in Berlin and 231 in Saxony.

19    For details regarding methods and research process see the methodological report (SVR-Forschungsbereich 2020b).

20    Offered answers were: "state policy/administration (e.g., state ministries)"; "federal policy/administration (e.g., federal ministries)"; "municipal administrations", "district administrations"; "other migrant organizations"; "welfare associations"; "churches and religious communities"; "other non-profit associations/organisations"; "commercial enterprises"; "foundations"; "universities and colleges"; "informal associations and initiatives"; "twin cities abroad"; "authorities/organisations abroad"; "others".

21    A minor importance in this context only refers to the quantitative share of MOs' collaborations. That does not mean that these cooperation partners cannot be of particular importance for individual MOs.

22    We restricted the number of activity fields the MOs could choose, because most MOs are multifunctional organizations, as shown above. To find out the differences between the MOs in their portfolio, we decided to survey only the five most important activities. The results confirm this decision. On average, the MOs in the survey mentioned 4.4 activity areas (SVR-Forschungsbereich 2020a, p. 26).

23    The activity fields were: "offers for women"; "anti-discrimination work"; "employment agency"; "exchange between people with and without migrant backgrounds"; "consulting"; "education"; "parental/family work"; "development cooperation"; "health"; "child and youth work"; "artistic-cultural activities (e.g., music, theatre)"; "cultivation of culture(s) of origin"; "cultivation/teaching of language of origin"; "political representation of interests"; "religion"; "senior citizens' work"; "sport"; "translations"; "environmental protection and nature conservation"; "support for refugees"; "representation of professional interests"; "science and research"; and "other".

24    In the main publication of this research project, we differentiated three MO-types via cluster analysis based on the self-designation of the surveyed MOs. Although these types had different foci in their work, it was established that many MOs were active in a lot of areas (SVR-Forschungsbereich 2020a, p. 39). This is also reflected in other studies. Studies of religious migrant communities,

for instance, show that many of these organizations also do social work for their members (Halm et al. 2012; Halm and Sauer 2015; Klöckner 2016, p. 269; Nagel 2016, p. 92).

[25] The number of employees includes all kinds of paid work in the organization, e.g., permanent employees as well as freelancers, so-called Minijobs or paid internships.

[26] MOs were asked if they received funding, donation, funding and donation, or none of them in 2019. We did not ask about the amount of funding or donation, to avoid risking an interview abort when faced with these sensitive questions, which is particularly likely in an online survey.

[27] To make sure that there is no strong multi-collinearity between the independent variables, which might influence the prediction, we checked the variance inflation factors (VIF) of predictor variables for all regression models. For the interpretation, we follow the conservative recommendation of Urban and Mayerl (2006, p. 232) and regard a VIF > 5 as an indication of collinearity problems. All VIF were below 2, so there was no sign for multi-collinearity.

[28] We also calculated models considering the income as a second predictor for financial resources. As this variable has many missing cases and we hardly could find effects, we excluded it from our analysis.

[29] As mentioned above, we did the same for the year of foundation. In our sample are few MOs that were founded in the 1950s and 1960s, which would have a strong influence on the average year of foundation.

[30] Cramer's V is $-0.14^*$ for the correlation between cultural/religious activities and social work, and $-0.24^*$ for the correlation between cultural/religious and political activities.

[31] Since the three activity fields represent three independent variables, testing for significance between the three fields is not directly possible.

[32] However, the explanatory power of this finding is limited, as our survey only asked about the number of unpaid volunteers, and not about how regularly and consistently they volunteer. This question is likely to be significant, as the three fields of action sometimes require different levels of commitment: In many areas of social work—and, to some extent, also in the cultural-religious sphere—voluntary commitment requires a certain constancy. In other words: it may be easier in the political sphere to get involved only once in a while. In the case of political activity, perhaps people were also counted who "only" designed or distributed a flyer once, while respondents of MOs with a clear focus on social work or culture/religion were more likely to have thought of regularly engaged people when answering the question. Even though this interpretation is speculative, it emphasizes a certain need for a more detailed revision of different ideas and profiles of unpaid volunteers in MOs in further research.

[33] This result can partly be explained with funding guidelines, because they often exclude religious projects and those of cultural maintenance (SVR-Forschungsbereich 2020a, p. 68).

[34] We use odds ratios to interpret the correlations. However, since this coefficient is not linearly related to the probabilities, we primarily interpret the direction of the correlation, instead of the effect size (Behnke 2015, p. 75).

[35] As the variable to measure the diversity of cooperation partners has been calculated on the basis of the different cooperation partners, this result does not deliver additional knowledge. Indeed, we see almost the same effects as in Table 2.

[36] We also calculated this interaction for the activity culture and religion. Paid employees appear to be less important in this area than for other activities. When paid employees and religious/cultural activities co-occur, there is a lower cooperation diversity than is the case for other activity foci.

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
