# Peer review of "Stronger Together? Determinants of Cooperation Patterns of Migrant Organizations in Germany"

_socsci, doi:10.3390/socsci12040223_

Round 1
Reviewer 1 Report
I consider the paper original and innovative. I highlight the originality, the very well planned introduction and the Putnam work to construct the theoretical framework.
However: please situate the method after the theoretical framework. Then,please situate the hypotheses and the analysis. Please be careful with the organization and structure of your work. The discussion are great!
Author Response
Thank you very much for your constructive comments. Based on your recommendations, we have revised the structure of section 2 once again. In section 2.1, we discuss theoretical assumptions and results of previous studies on the relationship between the content orientation of an MO and its likelihood of cooperation; in section 2.2, we address possible relationships between human and financial resources as well as other structural characteristics of the organizations and collaborations. In 2.3, as before, we formulate our hypotheses based on the considerations in 2.1 and 2.2. We hope that the structure of the article will become clearer in this way.
Reviewer 2 Report
This paper deals with an interesting topic - networks usage of migrants' associations from the perspective of Germany, which is one of the most important immigrants' destinations in the EU. Definitelly the strong side and potential contribution of the paper is the quantitative aspect of the study, which is a relative novelty. Nevetherless, the paper in its current form has potential flaws and weaknesses which should be adressed. 1. Country-specific context - I agree that the legal framework between migrants' associations do vary between destination countries, but there are also some basic functions of such organizations which are discussed in migration studies and studies on migration studies in general. Currently, the paper misses completelly the literature review of studies on migrant associations in Europe (see basic literature below) 2. Theoretical bacground poorly developed - the paper does not discuss the basic theories on how migrant associations are created, developed and transformed (see the literature below). As such, the paper now is purely explorative and does not test any existing theory on migrants' associations and on integration of immigrants. The hypotheses of the study are not grounded in any theory and seemed to be taken from no-where. 3. Sampling - why those 4 federal states were chosen? Due data avaliability or other factor? The response rate is slightly above 10% - not dramatic indicator, but have you done any comparison with respondents and non-respondents based on any major characteristics, excluding the geographical location (which you included already)? Probably looking on such variables as budget size, number of employees, and date of estabilishment could be useful to see whether your observations could be generalized for entire population. 4. Econometric specification - basically the results do not look bad, but maybe you can integrate at least some continuuous variables? For now, most of the ones you use are either dummies or ordinal ones. I wish you good luck with developing your paper: it has some flaws, but the overall impression/assesment is rather positive. Literature: Danese, G. (2001). Participation beyond citizenship: migrants' associations in Italy and Spain. Patterns of prejudice, 35(1), 69-89. Góis, P., & Marques, J. (2023). Migrant associations, other social networks of Portuguese diaspora, and the modern political engagement of non-resident citizens. European Political Science, 1-18. Odmalm, P. (2004). Civil society, migrant organisations and political parties: theoretical linkages and applications to the Swedish context. Journal of Ethnic and Migration studies, 30(3), 471-489. Morad, M., & Della Puppa, F. (2019). Bangladeshi migrant associations in Italy: transnational engagement, community formation and regional unity. Ethnic and Racial Studies, 42(10), 1788-1807. Amelina, A., & Faist, T. (2008). Turkish migrant associations in Germany: Between integration pressure and transnational linkages. Revue européenne des migrations internationales, 24(2), 91-120. Fauser, M. (2016). Migrants and cities: The accommodation of migrant organizations in Europe. Routledge. Breton, R. (1964). Institutional completeness of ethnic communities and the personal relations of immigrants. American journal of Sociology, 70(2), 193-205. Myrberg, G. (2013). Surveying migrants and migrant associations in Stockholm. Surveying ethnic minorities and immigrant populations: Methodological challenges and research strategies, 131-146. Gerstnerova, A. (2016). Migrant associations' dynamics in Luxembourg. Ethnicities, 16(3), 418-431. Gallo, E., & Sai, S. (2013). Should we talk about religion? Migrant associations, local politics and representations of religious diversity: the case of Sikh communities in central Italy. In Sites and Politics of Religious Diversity in Southern Europe (pp. 279-308). Brill.
Author Response
Thank you very much for your numerous constructive and critical comments and the literature recommendations, which certainly benefit the quality of the text. In the following, we address the four central points of criticism that you noted in your review.
1) Country-specific context: You are of course correct that some key conditions and characteristics of MOs are quite comparable despite country-specific differences. At the same time, we note that the issue of cooperation structures is not or only marginally considered in many studies. Therefore, especially in the introduction, we have referred to some publications on other countries that deal with the content orientation of migrant organizations. Where this was possible, we have also made corresponding additions in the second section. Nevertheless, we consider it useful to keep the clear focus on the cooperation structures of migrant organizations and not to focus on the question of the functions of MOs for integration processes.
2) Theoretical framework: From our point of view, especially the lack of theoretical reflection on the influence of resources and structural characteristics is a problem, since we rely on social capital theory for the effects of activity fields. Unfortunately, we find little in prominent approaches, such as political opportunity structures theory or resources mobilization theory, that address effects on cooperation. In this respect, hypotheses on the influence of resources on cooperation unfortunately still rely heavily on empirical results of previous studies. However, we made a link to resource mobilization theory with respect to the relationships between resources and activity fields, which we discuss at the end of the second section. Furthermore, we refer again to the considerations made in section 2.1 and 2.2 when formulating the hypotheses to establish a clearer link between section 2.1 and 2.2 and our hypothesis.
3) Sampling: Section 3 was supplemented by corresponding explanations on the selection of the federal states and the sample.
4) Econometric specification: The variables “number of employees”, “number of volunteers”, and “year of foundation” are continuous variables in the data set, which we categorized for the regression models, because they are unfortunately skewed distributed and therefore cannot be used as metric predictors in the analyses. There was only a very brief reference to this in the description of the results. This has been placed a little more clearly in the methods section 3 with a more detailed explanation.
Thank you again for your comments, we hope we have implemented them appropriately.
Round 2
Reviewer 2 Report
I have read the new version of the manuscript and the response file. I can say the authors made substantial improvements especially with the description on how data was collected, and to a lesser extent on the contribution of the study. I guess the literature review could be still more precise, but I can see you rushed really for a re-submission. Honestly, the quality major revision cannot be done in such a quick pace.
Summing up: I am a bit dissapointed because you could have worked harder to make a better improvement of the paper. The paper had really nice potential for making a real contribution to the literature. Yet, with all of these time pressure from mdpi I guess some improvement was made. So I can say I am not an enthusiast of this paper in its current form, but I will not oppose any longer to its publication.
Author Response
Dear reviewer,
thank you again for your critical feedback. Based on your initial recommendations, we just tried in this second revision to link our research question once again a little better to international research, as far as the given time has allowed. Accordingly, we have made a few additions, although the basic framework has not been changed. Nevertheless, we hope that the new revision will contribute to an improvement of the paper.
Best regards